# Reliability of Gridded Precipitation Products for Water Management Studies: The Case of the Ankavia River Basin in Madagascar

Zonirina Ramahaimandimby [1,*], Alain Randriamaherisoa [2], François Jonard [1,3], Marnik Vanclooster [1] and Charles L. Bielders [1]

1   Earth and Life Institute, Université Catholique de Louvain, 1348 Ottignies-Louvain-la-Neuve, Belgium
2   Civil Engineering Department, University of Antananarivo, BP 1500, Antananarivo 101, Madagascar
3   Earth Observation and Ecosystem Modelling Laboratory, University of Liège, 4000 Liège, Belgium
*   Correspondence: zoramahaimandimby@gmail.com; Tel.: +261-34-92-345-45

**Abstract:** Hydrological modeling for water management in large watersheds requires accurate spatially-distributed rainfall time series. In case of low coverage density of ground-based measurements, gridded precipitation products (GPPs) from merged satellite-/gauge-/model-based rainfall products constitute an attractive alternative. The quality of which must, nevertheless, be verified. The objective of this study was to evaluate, at different time scales, the reliability of 6 GPPs against a 2-year record from a network of 14 rainfall gauges located in the Ankavia catchment (Madagascar). The GPPs considered in this study are the African Rainfall Estimate Climatology (ARC2), the Climate Hazards Group Infrared Precipitation with Station data (CHIRPS), the European Centre Medium-Range Weather Forecasts ECMWF Reanalysis on global land surface (ERA5-Land), the Integrated Multi-satellitE Retrievals for Global Precipitation Measurement V06 Final (IMERG), the Precipitation Estimation from Remotely Sensed Information using Artificial Neural Networks Cloud Classification System (PERSIANN-CCS), and the African Rainfall Estimation (RFEv2) products. The results suggest that IMERG ($R^2 = 0.63$, slope of linear regression $a = 0.96$, root mean square error RMSE = 12 mm/day, mean absolute error MAE = 5.5 mm/day) outperforms other GPPs at the daily scale, followed by RFEv2 ($R^2 = 0.41$, $a = 0.94$, RMSE = 15 mm/day, MAE = 6 mm/day) and ARC2 ($R^2 = 0.30$, $a = 0.88$, RMSE = 16 mm/day, MAE = 6.7 mm/day). All GPPs, with the exception of the ERA5, overestimate the 'no rain' class (0–0.2 mm/day). ARC2, IMERG, PERSIANN, and RFEv2 all underestimate rainfall occurrence in the 0.2–150 mm/day rainfall range, whilst CHIRPS and ERA5 overestimate it. Only CHIRPS and PERSIANN could estimate extreme rainfall (>150 mm/day) satisfactorily. According to the Critical Success Index (CSI) categorical statistical measure, IMERG performs quite well in detecting rain events in the range of 2–100 mm/day, whereas PERSIANN outperforms IMERG for rain events larger than 150 mm/day. Because it performs best at daily scale, only IMERG was evaluated for time scales other than daily. At the yearly and monthly time scales, the performance is good with $R^2 = 0.97$ and 0.87, respectively. At the event time scale, the probability distribution function PDF of rain gauge values and IMERG data show good agreement. However, at an hourly time scale, the correlation between ground-based measurements and IMERG data becomes poor ($R^2 = 0.20$). Overall, the IMERG product can be regarded as the most reliable gridded precipitation source at monthly, daily, and event time scales for hydrological applications in the study area, but the poor agreement at hourly time scale and the inability to detect extreme rainfall >100 mm/day may, nevertheless, restrict its use.

**Keywords:** Madagascar; GIRE SAVA; Ankavia; gridded precipitation products; IMERG

## 1. Introduction

Accurate precipitation data are essential for numerous theoretical and practical applications, be it for water balance calculations, flood warnings, drought monitoring, or water

resource management [1–3]. When properly installed and maintained, rain gauge observations provide accurate point-based precipitation measurements [4,5]. However, in the case of low coverage density, they are poorly adapted to deal with the high spatiotemporal heterogeneity in precipitation. The latter can result in large errors when rain gauge data are interpolated to larger scales, particularly in mountainous areas with complex terrain [6,7]. Furthermore, the spatial distribution of rain gauges is often highly uneven in practice, with few gauges in remote areas, in less developed regions, in areas with complicated terrain, or in forested areas [8]. As a result, in situ rain gauge data seldom matches the needs of applications that require precipitation data with high spatiotemporal resolution [8,9]. This is particularly true across vast swaths of the African continent [10].

As opposed to rain gauges, gridded precipitation products (GPPs) from merged satellite-/gauge-/model-based rainfall products have the advantage of offering wide spatial coverage [1,11]. There are currently a number of GPPs available, including ARC2 (African Rainfall Estimate Climatology version 2), CHIRPS (Climate Hazards Group Infrared Precipitation with Station Data), ERA5 (European Centre Medium-Range Weather Forecasts Reanalysis), IMERG v06 Final (Integrated Multi-satellitE Retrievals for Global Precipitation Measurement), PERSIANN-CCS (Precipitation Estimation from Remotely Sensed Information using Artificial Neural Networks Cloud Classification System), and RFEv2 (African Rainfall Estimation version 2), among others [1]. Recent GPPs also provide adequate spatial ($\leq 0.1 \times 0.1°$) and temporal (daily to sub-hourly, depending on the product) resolution, allowing for credible precipitation estimates in data-scarce environments or at ungauged locations [1,12,13]. They have been used in a variety of applications, including hydrological modeling, extreme event analysis, infrastructure design (based on rainfall frequency analysis), and water resource management [8,14,15].

Since GPPs are based on indirect rainfall estimation methods, the results will be subject to uncertainty due to measurement errors, sampling, retrieval methods, and bias correction processes [16,17]. The errors depend on the number and type of sensors taking measurements across a certain location at a given time, as well as the strategies used to assimilate the available data into a coherent gridded dataset [18,19]. Furthermore, the error characteristics differ based on the type of storm system, location, topography, and cloud properties [19]. Therefore, the accuracy of GPPs must be thoroughly explored both in time and space [20,21], and quantitative statistical evaluations are useful tools for assessing GPP precision [20,22]. Whereas some researchers assess GPPs based on the accuracy of streamflow rate predictions within hydrological modeling frameworks [22,23], most studies evaluate GPPs against gauge data or against estimates from ground-based weather radars [9,24].

Various studies have been undertaken to assess GPP performance at the global, continental, and regional levels during the last few decades. TRMM Multi-Satellite Precipitation Analysis (TMPA) products, for example, have been assessed in various parts of Africa, and the results revealed that TMPA products provide effective data in most regions [6,25]. In [16], the authors found that TMPA was the best product at a daily time scale over different parts of Central Africa. Following that success, ref. [20] proved that the Integrated Multi-satellite Retrievals for GPM (Global Precipitation Measurement), which integrates observations from many satellites of the GPM satellite constellation, improves the quality and spatiotemporal resolution of precipitation data. Other investigations in eastern Africa (Zimbabwe) show that ARC2 and RFEv2 estimate the precipitation gauge data better than other GPPs [17]. In addition, an evaluation conducted by [25] in equatorial and eastern Africa showed that IMERG performed better for daily scales, while CHIRPS outperformed other products at monthly and annual scales. Overall, the reliability of GPPs appears to be governed by a number of factors, including the study scale, location, time scale, and, most significantly, the availability of ground-based data used for calibration [17,20].

Despite the significant efforts undertaken so far to evaluate GPPs, those products continue to require extensive validation against ground observations in order to assess their quality and quantify the appropriate level of confidence in their use for various hydrological

applications [1]. Furthermore, ref. [26] highlights that the choice of GPP has a significant impact on runoff estimation and underlines the need for rigorous assessment with in situ observations to improve their confident application in water cycle research. As a result, temporal aspects and spatial distributions must be quantitatively analyzed. Nonetheless, the scale discrepancy problem persists when rain gauge data is used for validation. So far, the majority of existing GPP validation efforts in Africa have been conducted at large scales (country level or greater), with rain gauges separated by very large distances [8,17,27], and they were often performed on public datasets or GPCP-1DD (Global Precipitation Climatology Project One-Degree Daily Precipitation) data [28]. Since numerous water management issues are dealt with at smaller scales, it is therefore of high interest to also evaluate the ability of GPPs to capture rainfall variations across short distances (5–10 km) for applications in medium to large watersheds (i.e., ~$10^3$ km$^2$) [29].

The aim of this study was, therefore, to evaluate, at different time scales (hourly to yearly), the reliability of six major GPPs (ARC2, CHIRPS, ERA5-Land (hereafter ERA5), IMERG v06 Final (hereafter IMERG), RFEv2, and PERSIANN-CCS (hereafter PERSIANN)) for water management applications in medium-size watersheds in Africa. More specifically, GPP data were evaluated against a network of rain gauges installed in the Ankavia watershed (1116 km$^2$) in northeastern Madagascar. Water-related issues abound in Madagascar, strongly impacting economic development and environmental conservation [30,31]. Indeed, northeastern Madagascar is characterized by heavy rainfall (1500 to 2500 mm/year), caused by southeasterly exchanges that start in the Indian Ocean anticyclone and reach the highlands of the east [32]. This, along with deforestation from slash and burn, logging, and firewood harvesting, contributes to some of the world's greatest levels of erosion and catastrophic flooding [33,34]. Furthermore, as a result of climate change, more powerful cyclones and increasing sea levels directly threaten coastal settlements and exacerbate floods and erosion in coastal areas [35]. In contrast, during the dry season, some rivers in the north tend to dry up, and alternative groundwater sources are not always available [32]. Previous research has shown that pressure on water resources in Madagascar is increasing [36,37]. Hydrometeorological data are scarce and not always routinely collected, which impedes decision-making for integrated water resources management (IWRM), particularly at the basin scale [38,39]. Hence the use of reliable GPP seems unavoidable for hydrological modeling, drought monitoring, and water resources management.

## 2. Materials and Methods

### 2.1. Study Area

This study focuses on the Ankavia watershed, located between 14°50′–15°20′S and 49°50′–50°20′E in the SAVA region in northeast Madagascar (Figure 1). At a regional scale, the climate is governed by the southeasterly trade winds that originate from the Indian Ocean anticyclone, a zone of high atmospheric pressure that seasonally changes its position over the ocean [38]. The northeastern coast of Madagascar is most directly exposed to the trade winds and has the highest rainfall in the country [40]. Furthermore, the region is regularly affected by tropical storms and cyclones [31]. The area has a subequatorial climate with two main seasons; the hot, rainy season extends from November to April (approximately 70% of total annual precipitation), and the cooler, drier season from May to October. Temperatures range from 18 °C to 31 °C [41].

The hydrographic network in the SAVA region is dense and highly branched [42]. The majority of the rivers originate in mountainous massifs and flow eastward into the Indian Ocean. These rivers are heavily fed throughout the year, with low flows in October and November. Floods are common during the rainy season and are often exacerbated in the coastal zone by sediment accumulation [43]. For the last 60 years, the climatic data for the region have been provided solely by the Antalaha weather station located close to the coast (Figure 1).

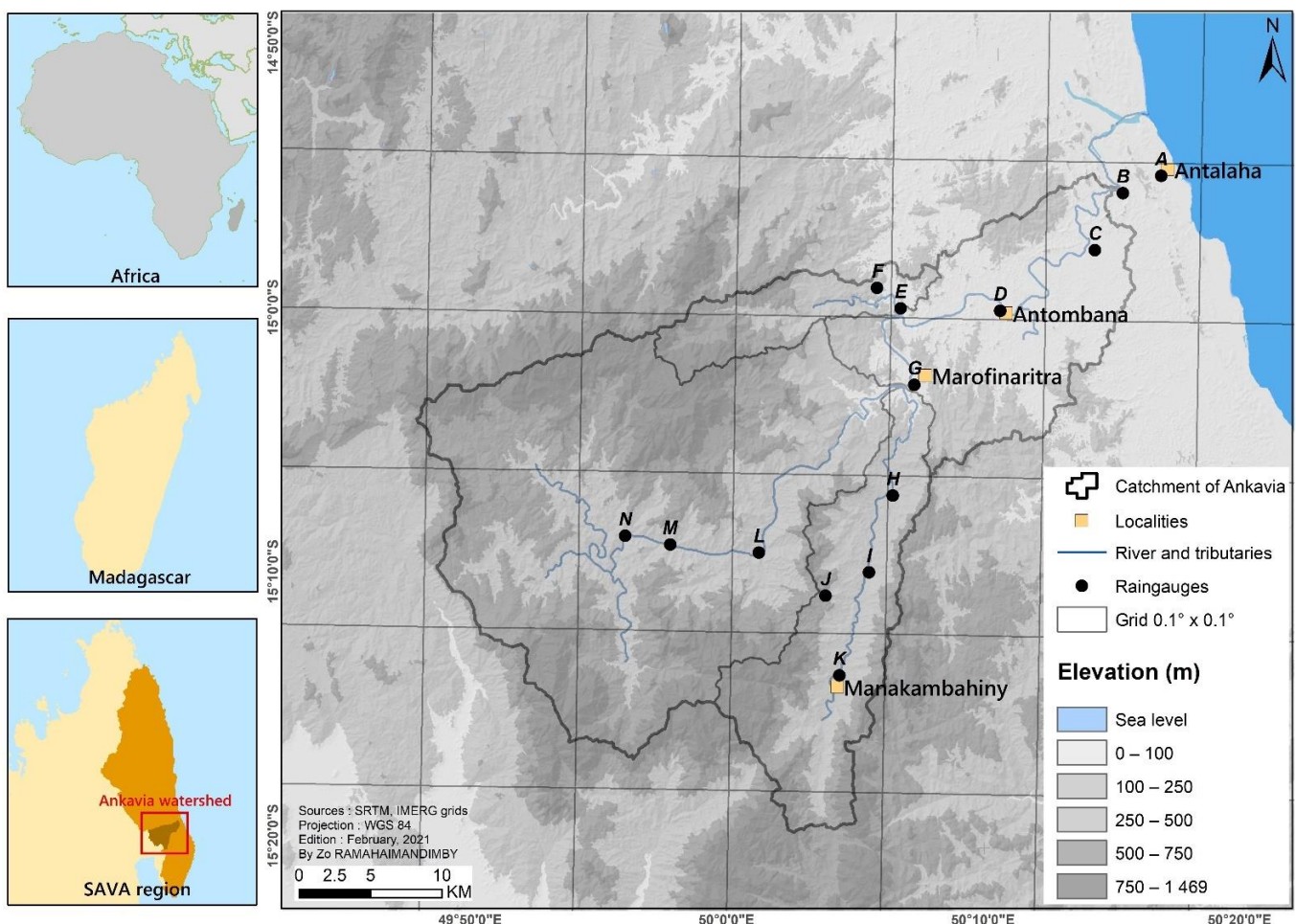

**Figure 1.** Location of the Ankavia watershed in northeastern Madagascar with rain gauge network (rain gauges numbered A to N).

The Ankavia watershed covers an area of 1116 km$^2$, i.e., roughly 5% of the total area of the SAVA region (Figure 1). It was chosen for this study due to its natural and social context, both exerting a strain on water resources. In particular, the Ankavia river provides water to the city of Antalaha (150,000 people in 2017). Altitude in the catchment varies from 14 m a.m.s.l. near the outlet in the east (hilly topography) to 1469 m a.m.s.l. in the southwest (mountainous topography). The western part of the Ankavia watershed is mainly occupied by primary forests, whereas the east is composed mostly of mosaic vegetation, including shrubs and herbaceous cover [44]. The vast majority of inhabited and cultivated areas are clustered around the major rivers [41].

### 2.2. Ground-Based Precipitation Data

Fourteen rain gauges and one meteorological station were established in the Ankavia catchment as part of the GIRE SAVA project (Gestion Integrée des Ressources en Eau in the SAVA region) (Figure 1). The rain gauge at the Marofinaritra climate station is a Campbell Scientific® ARG100, whereas the other 13 rain gauges scattered within the basin are HOBO® RG3-M instruments. These gauges are not part of the Global Telecommunications System (GTS) network. They are set with a recording interval of one hour. The stations are positioned 1.5 m above the ground, and their elevation ranges from 25 m to 663 m a.m.s.l., with the majority of the stations located along the rivers at low and mid altitudes and 90% of the rain gauges located below 300 m (Figure 1). Because of the remoteness, dense vegetation, and lack of roads, as well as the difficulty of ensuring routine maintenance, no rain gauge could be installed in the high-elevation mountainous region (>1000 m a.m.s.l). The rainfall

data used in the current study were collected for a two-year period, from September 2018 to August 2020, thanks to regular monthly maintenance and data collection.

### 2.3. Gridded Precipitation Products

Six GPPs were used in this study to compare with observed rain gauge data (Table 1). These products were chosen based on the availability of recent time series (from 2018 onwards), spatial ($\leq 0.1°$) and temporal ($\leq$daily) resolutions that make them suitable for hydrological applications at the scale of the Ankavia watershed, near-real-time availability, public domain, and their coverage of Africa.

### 2.3.1. ARC2

ARC2 was developed by the National Oceanic and Atmospheric Administration (NOAA) Climate Prediction Center (CPC), which offers daily rainfall data for Africa [1]. It uses inputs from two sources: (i) 3-hourly geostationary infrared (IR) data centered over Africa from the European Organization for the Exploitation of Meteorological Satellites (EUMETSAT), and (ii) quality-controlled GTS gauge observations reporting 24 h rainfall accumulations across Africa [45]. ARC2 has a spatial resolution of 0.1° over Africa (40°N–40°S, 20°W–55°E) with daily temporal resolution and can be downloaded at: https://iridl.ldeo.columbia.edu/SOURCES/.NOAA/.NCEP/.CPC/.FEWS/.Africa/.DAILY/.ARC2/.daily/ (accessed on 3 June 2021).

### 2.3.2. CHIRPS

Since 1981, the CHIRPS product has provided daily precipitation data with a spatial resolution of 0.05° for a quasi-global coverage of 50°N–50°S [12]. The most recent product is Version 2.0, which was released in February 2015. The CHIRPS product and associated data can be found at: https://climateserv.servirglobal.net (accessed on 3 June 2021). The main datasets used for the construction of the CHIRPS product are the monthly precipitation climatology (CHPclim) information based on thermal infrared data archived from CPC and NOAA National Climate Data Center (NCDC), the Version 7 TRMM 3B42 data, the Version 2 atmospheric model rainfall field from the NOAA Climate Forecast System (CFS), and rain gauge stations [46]. First, the cold cloud duration (CCD) data are calibrated with TRMM 3B42 to generate 5-day CCD-based precipitation estimates, which are then converted to fractions of long-term mean precipitation estimates [46,47]. The fractions are then multiplied by CHPclim data to remove systematic bias, and the resulting product is known as the CHIRP product [28]. Finally, the CHIRP product is combined with data from rain gauge stations using a modified inverse distance weighting algorithm to generate the CHIRPS [46]. All of the preceding processing is carried out on a 5-daily basis. Using a simple redistribution method, the CCD and CFS data are finally used to disaggregate the 5-daily products to daily precipitation estimates [47].

### 2.3.3. ERA5

ERA5 is the most recent edition of the global atmospheric reanalysis of the ECMWF from 1979 [48]. ERA5-Land was created by rerunning the land component of the ERA5 climate reanalysis and spans the same time period as ERA5, from January 1950 to near real-time (NRT) [49]. ERA5-Land is generated in a single simulation that is not coupled to the atmospheric module of the ECMWF's Integrated Forecast System. Observations have an indirect effect on the simulation due to the atmospheric forcing of ERA5 [48,50]. This forcing is used to drive the single ERA5-Land simulation and was obtained by integrating observations using a 4D-Var data assimilation system and a Simplified Extended Kalman Filter [51]. The fields are overlain for all oceans and have an hourly resolution. The Climate Data Store (CDS) Climate Copernicus website was used to download hourly total precipitation for the study period with a spatial resolution of approximately 0.1 × 0.1° (available at https://cds.climate.copernicus.eu/cdsapp#!/dataset/reanalysis-era5-land?tab=overview) (accessed on 3 June 2021).

### 2.3.4. IMERG

IMERG is available at https://giovanni.gsfc.nasa.gov/giovanni/ (accessed on 3 June 2021). The GPM project, a collaboration between the National Aeronautics and Space Administration (NASA) of the United States and the Japan Aerospace Exploration Agency (JAXA), began in 2014 to provide half-hourly global precipitation data with a spatial resolution of 0.1° [52]. The GPM satellite is equipped with two major sensors: the GPM Microwave Imager (GMI), which measures precipitation intensity, depth, and duration, and the Dual-frequency Precipitation Radar (DPR), which observes storm internal structure within and beneath clouds [53]. The GPM Constellation provides three levels of data processing (IMERG products), but the most commonly used are the gridded products that combine GMI and DPR rainfall averages or rainfall estimates combined from data of all active and passive microwave instruments in the GPM Constellation [52,54]. There are three daily IMERG products: IMERG Early Run (near real-time with a latency of 4 h), IMERG Late Run (reprocessed near real-time with a latency of 14 h), and IMERG Final Run (gauged-adjusted with a 4-month latency) [54]. In this study, we selected the IMERG-v06 Final Run half-hourly products [53].

### 2.3.5. PERSIANN

The PERSIANN provides hourly precipitation estimates at the spatial resolution of 0.04° for the quasi-global coverage of 60°N–60°S from 2003 to the present [55]. PERSIANN, developed at the University of Arizona and now operated by the Center for Hydrometeorology and Remote Sensing (CHRS) at the University of California Irvine (UCI), is based on an adaptive Artificial Neural Network (ANN) model that estimates fine-scale precipitation distribution using IR information (10.7 μm) from geostationary satellites in analyzing local and regional cloud properties [56]. The PERSIANN-CCS algorithm converts rain rates from satellite cloud images in several steps, which consist of: (i) separating cloud images into recognizable cloud patches, (ii) extracting cloud properties based on their shape, texture, and coldness, (iii) clustering cloud patches into orderly subgroups, and (iv) calibrating cloud-top temperature and rainfall ($T_b$-$R$) correlations for the various cloud categories using gauge-corrected radar hourly rainfall data [55]. The product is available at: http://chrsdata.eng.uci.edu/ (accessed on 3 June 2021).

**Table 1.** Selected gridded precipitation products.

| Dataset | Full Name | Spatial Resolution | Timescale (Highest Resolution) | Period of Availability | Reference |
|---|---|---|---|---|---|
| ARC2 | African Rainfall Climatology version 2 | 0.10° | Daily | 1983–Present | [45] |
| CHIRPS | Climate Hazards Group Infrared Precipitation with station data | 0.05° | Daily | 1981–Present | [47] |
| ERA5 | ECMWF Reanalysis version 5 on global land surface | 0.10° | Hourly | 1979–Present | [50] |
| IMERG | Integrated Multi-satellitE Retrievals for Global Precipitation Measurement | 0.10° | Half-hourly | 2000–Present | [53] |
| PERSIANN | Precipitation Estimation from Remotely Sensed Information using Artificial Neural Networks—Cloud Classification System | 0.04° | Hourly | 2003–Present | [55] |
| RFEv2 | Climate Prediction Center (CPC) African Rainfall Estimates version 2 | 0.10° | Daily | 2001–Present | [57] |

### 2.3.6. RFEv2

Finally, RFEv2 is produced by the NOAA-CPC. It is primarily designed for the Famine Early Warning Systems Network to aid in disaster monitoring across Africa [7]. The product estimates daily precipitation for Africa with a spatial resolution of 0.1°. RFEv2 receives data from four operational sources: (1) daily GTS rain-gauge data, (2) Advanced Microwave Sounding Unit (AMSU)-based rainfall estimates, (3) Special Sensor Microwave Imager/Sounder (SSMIS)-based estimates, and (4) the Geostationary Operational Environmental Satellite (GOES) precipitation index (GPI) calculated from cloud-top infrared (IR) temperatures on a half-hourly basis [57]. It is available at https://iridl.ldeo.columbia.edu/SOURCES/.NOAA/.NCEP/.CPC/.FEWS/.Africa/.DAILY/.RFEv2// (accessed on 9 June 2021). However, the use of polar-orbiting Passive Microwave (PM) and geostationary IR data differs between ARC and RFE. ARC uses 3-hourly IR data rather than 30-min data and does not include PM estimates, whereas RFE does [45,57].

### 2.4. Data Comparison Methodology

### 2.4.1. Data Quality Control

Initially, the gridded product data were downloaded in files with half-hourly, hourly, or daily time steps (depending on availability; Table 1) based on the GMT 0:00 time zone. Negative values were removed from the gridded datasets. When necessary, daily totals were generated by summing the half-hourly or hourly files. For the study area (Madagascar +03H00 GMT), the observed rain gauge time series data were adjusted to the GMT and aggregated at an hourly or daily scale to be compared with the GPPs.

### 2.4.2. Data Processing

Because of the scale discrepancy between GPPs and rain gauge data (Figure 2), two approaches were used to assess the performance of GPPs: (i) point-to-grid, (ii) point-gridded approach. Point-to-grid compares the precipitation recorded at each rain gauge with the precipitation from the GPP grid (0.04°, 0.05°, and 0.1°, respectively) that encompasses the rain gauge (Figure 2a). Because the location of rain gauges most often does not coincide with GPP grid centroids (Figure 2a), a second strategy was implemented: the point-gridded approach. In practice, a cell is delineated around each rain gauge (cell size of 0.04, 0.05, or 0.1° depending on the GPP; Table 1). Then, the rainfall value in those new cells was estimated as the area-weighted mean (max. 4) of the GPP grid cells overlapping with the new cell (Figure 2b). A third approach could have been to define an 'area of influence' (Thiessen polygon) around each rain gauge and calculate area-weighted averages for each grid to be compared with the gridded data ('grid-to-grid' approach). However, such an approach leads to very different situations from one grid cell to another, some grids being in the area of influence of a single rain gauge while other grids are in the area of influence of up to five rain gauges. In addition, because the spatial resolution varies across GPPs, such an approach may introduce bias in the comparison.

### 2.4.3. Rainfall Event Definition and Properties

In addition to evaluation at daily and hourly time scales, the most performing GPP was evaluated at the event time scale. There are numerous methods for identifying individual rainfall events [58]. In this study, based on a study conducted by [59] over a tropical area (Brazil), a minimum inter-event time interval of 6 h and a minimum rainfall depth threshold of 2.5 mm were chosen for the evaluation. In other words, a cumulative rainfall depth > 2.5 mm is required to be considered as a rainfall event. The temporal resolution used to define rainfall events is one hour for both rain gauge and GPP data.

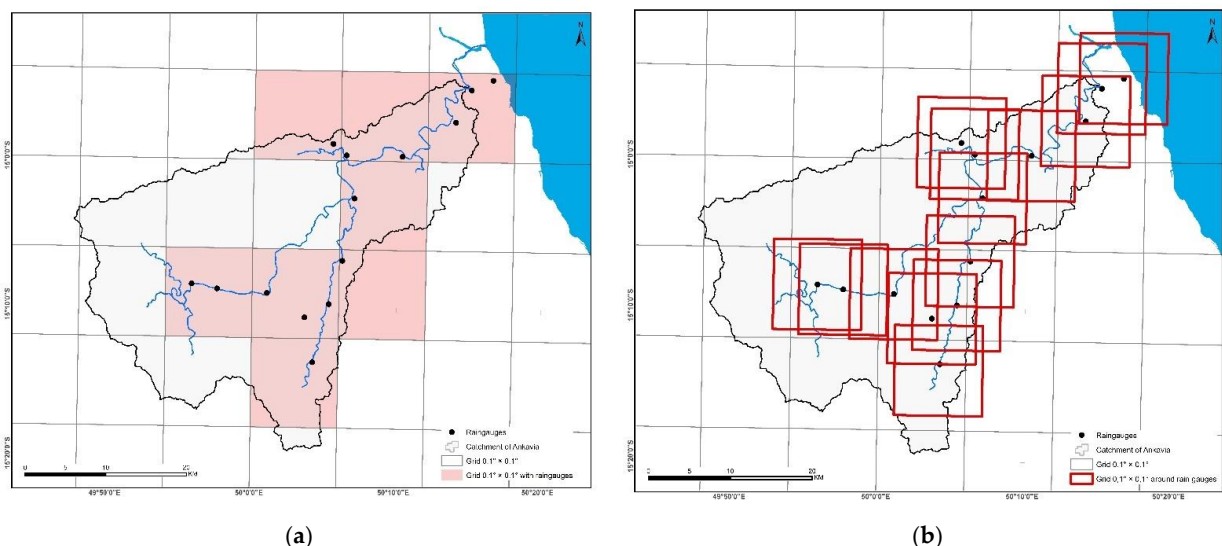

(**a**)  (**b**)

**Figure 2.** Data processing with two different approaches: (**a**) point-to-grid, (**b**) point-gridded.

### 2.4.4. Metrics for Accuracy Assessment

Several widely used statistical indices (Table 2) were adopted to quantify the performance of the six GPPs against rain gauge observations: Coefficient of determination ($R^2$), Slope of the linear regression (a), Root Mean Square Error (RMSE), and Mean Absolute Error (MAE).

**Table 2.** Statistical metrics used to quantify GPP performance.

| Name/Symbol | Formula | Optimal Value |
|---|---|---|
| Coefficient of determination/$R^2$ | $R^2 = (CC)^2 = \left( \dfrac{\sum_{i=1}^{n}(0i-\bar{0})(Pi-\bar{P})}{\sqrt{\sum_{i=1}^{n}(Oi-\bar{O})^2 \sum_{i=1}^{n}(Pi-\bar{P})^2}} \right)^2$ | 1 |
| Slope of linear regression/a | $Y = aX + b$ | 1 |
| Root Mean Square Error/RMSE | $RMSE = \sqrt{\dfrac{\sum_{i=1}^{n}(Pi-Oi)^2}{n}}$ | 0 |
| Mean Absolute Error/MAE | $MAE = \dfrac{1}{n} * \sum_{i=1}^{n} |Oi - Pi|$ | 0 |
| **Categorical statistical metrics** | | |
| Probability of Detection/POD | $POD = \dfrac{Hits}{Hits+Misses}$ | 1 |
| False Alarm Ratio/FAR | $FAR = \dfrac{False\,Alarm}{Hits+False\,Alarm}$ | 0 |
| Critical Success Index/CSI | $CSI = \dfrac{Hits}{Hits+False\,Alarm+Misses}$ | 1 |

P = Gridded Products value, O = Observed (rain gauge) value, $\bar{P}$ = average value of P, $\bar{0}$ = average value of O, $n$ = number of samples, X is the explanatory variable (O), Y is the dependent variable (P). The coefficient of determination $R^2$ was computed using linear fit. *Hits* denotes the number of observed precipitation occurrences correctly detected by the gridded products. *Misses* represents the number of precipitation occurrences observed by the rain gauges but not detected by the gridded products. *False Alarm* indicates the number of precipitation occurrences not observed by the rain gauges but detected by the gridded products.

In addition, we also evaluated the capability of the GPPs in reproducing the distribution of observed precipitation intensities using the Probability Distribution Function (PDF) of daily rainfall intensities. For this purpose, we categorized precipitation into 12 different classes: 0–0.2 mm/day, 0.2–0.5 mm/day, 0.5–1 mm/day, 1–2 mm/day, 2–5 mm/day, 5–10 mm/day, 10–20 mm/day, 20–50 mm/day, 50–100 mm/day, 100–150 mm/day, 150–200 mm/day, and >200 mm/day.

Finally, the probability of detection (POD), false alarm ratio (FAR), and critical success index (CSI) were calculated to evaluate the precipitation detection ability of the six

GPPs (Table 2). These indices aim at evaluating whether the estimated daily precipitation coincides with the precipitation observed on the ground. Specifically, POD represents the fraction of observed precipitation occurrences correctly detected by a given GPP. FAR corresponds to the fraction of detected precipitation occurrences that are incorrectly detected by a given GPP, while CSI measures the overall fraction of (detected and observed) precipitation occurrences correctly detected by a given GPP. The POD, FAR, and CSI values all range between 0 and 1. POD and CSI have perfect scores of 1, while FAR has a perfect score of 0. These indices are calculated for the different daily rainfall classes defined above.

## 3. Results

### 3.1. Overall GPP Performance at Daily Time-Scale

The comparison between rain gauge measurements and gridded rainfall product estimates reveals large differences between the six GPPs (Figure 3). However, all statistical metrics follow a similar pattern. Overall, IMERG data correlate best with the rain gauge data: highest $R^2$, the slope of the regression closest to 1, lowest RMSE and MAE. ARC2, PERSIANN, and RFEv2 perform rather similarly, though RFEv2 tends to have a better slope and MAE and PERSIANN a better $R^2$ than the two other products. ERA5 and, especially, CHIRPS perform worst.

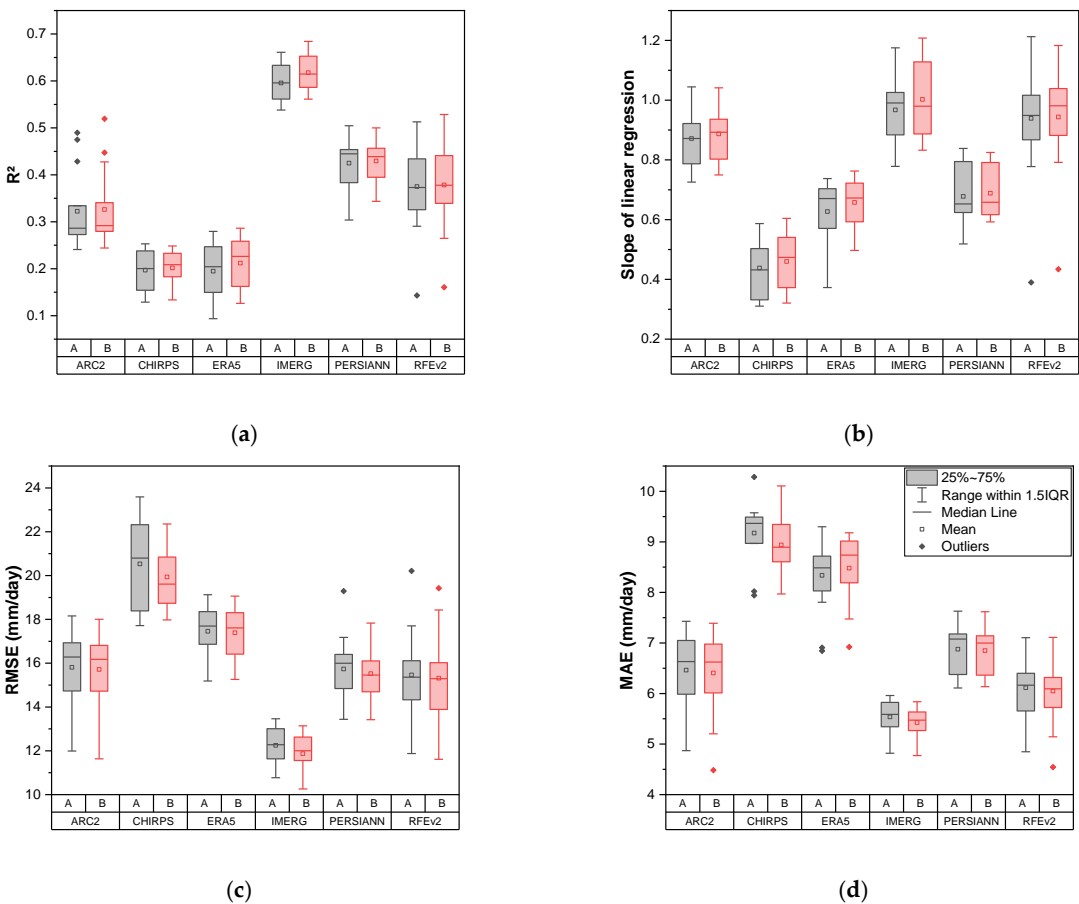

**Figure 3.** Comparison of ground-based precipitation data with 6 GPPs at a daily time scale from September 2018 to August 2020 based on 4 indicators: (**a**) Coefficient of determination, (**b**) Slope of linear regression, (**c**) Root Mean Square Error, (**d**) Mean Absolute Error. Each box plot is based on 2 years of data from 14 rain gauges in the Ankavia watershed. 'A' refers to the point-to-grid approach (grey boxes and whiskers) and 'B' to the point-gridded approach (red boxes and whiskers) (see Figure 2). Box edges correspond to the 25th (Q1) and 75th (Q3) percentiles. Whiskers extend to Q1-1.5IQR (lower bound) and Q3+1.5IQR (upper bound), with IQR = Q3–Q1 (Inter-Quartile Range). Points outside the box are outliers.

Overall, the statistical indices are generally similar or slightly better using the point-gridded approach compared to the point-to-grid approach (Figure 3). As a result, the point-gridded approach was used for the remainder of the analyses.

### 3.2. GPP Performance at Daily Time Scale across the Watershed

Figure 4 displays the $R^2$, slope, RMSE, and MAE for each rainfall gauge at a daily time scale across the Ankavia watershed. The greener the circle is, the closer the indicator is to its optimum value. In contrast, the color red indicates poor performance. Overall, IMERG shows good and fairly uniform levels of agreement across the entire watershed for all four indices. The performance of PERSIANN, RFEv2, and ARC2 varies widely from one location to another. CHIRPS and ERA5 show the poorest performance across the entire watershed.

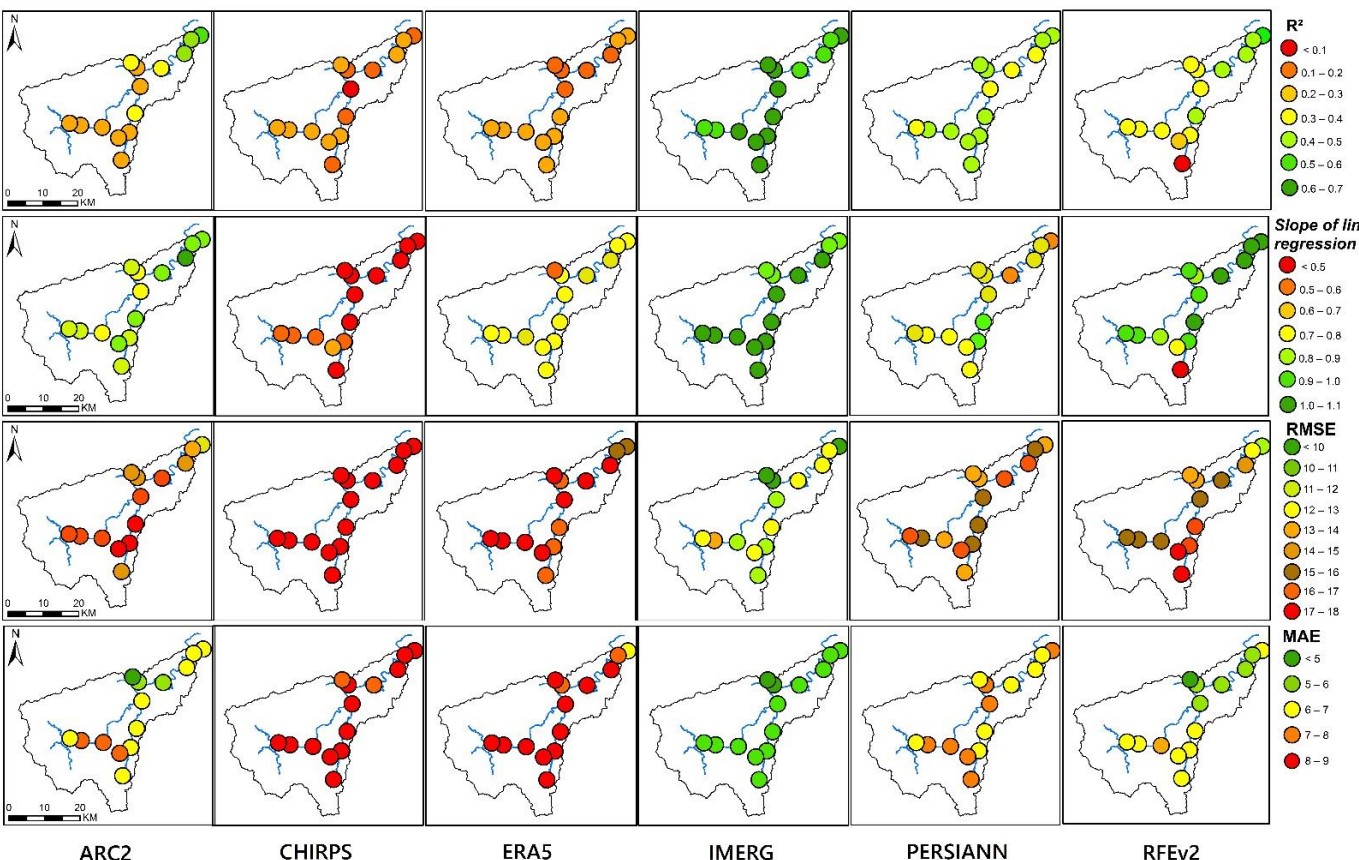

**Figure 4.** Spatial distribution of the statistical evaluation metrics across the Ankavia watershed based on the point-gridded approach at a daily time scale from September 2018 to August 2020.

### 3.3. Daily Rainfall Probability Distribution Function

PDFs computed from the six GPPs and the rain gauge data over the Ankavia catchment are shown in Figure 5. Overall, all GPPs' PDFs follow the same general trend as the rain gauge PDF, except for ERA5 and CHIRPS for specific ranges. ARC2, CHIRPS, IMERG, PERSIANN, and RFEv2 tend to overestimate the precipitation class between 0–0.2 mm/day. Most GPPs tend to underestimate the precipitation class between 0.5 to 10 mm/day, while ERA5 overestimates frequency in that precipitation range. Furthermore, most GPPs tend to underestimate the frequencies in precipitation classes > 150 mm/day. More specifically, ARC2 cannot retrieve precipitation events > 150 mm/day. IMERG cannot retrieve precipitation events > 200 mm/day, while ERA5 and RFEv2 can retrieve it but strongly underestimate this class. Only CHIRPS and PERSIANN perform well for rainfall > 200 mm/day, although they underestimate this rainfall range.

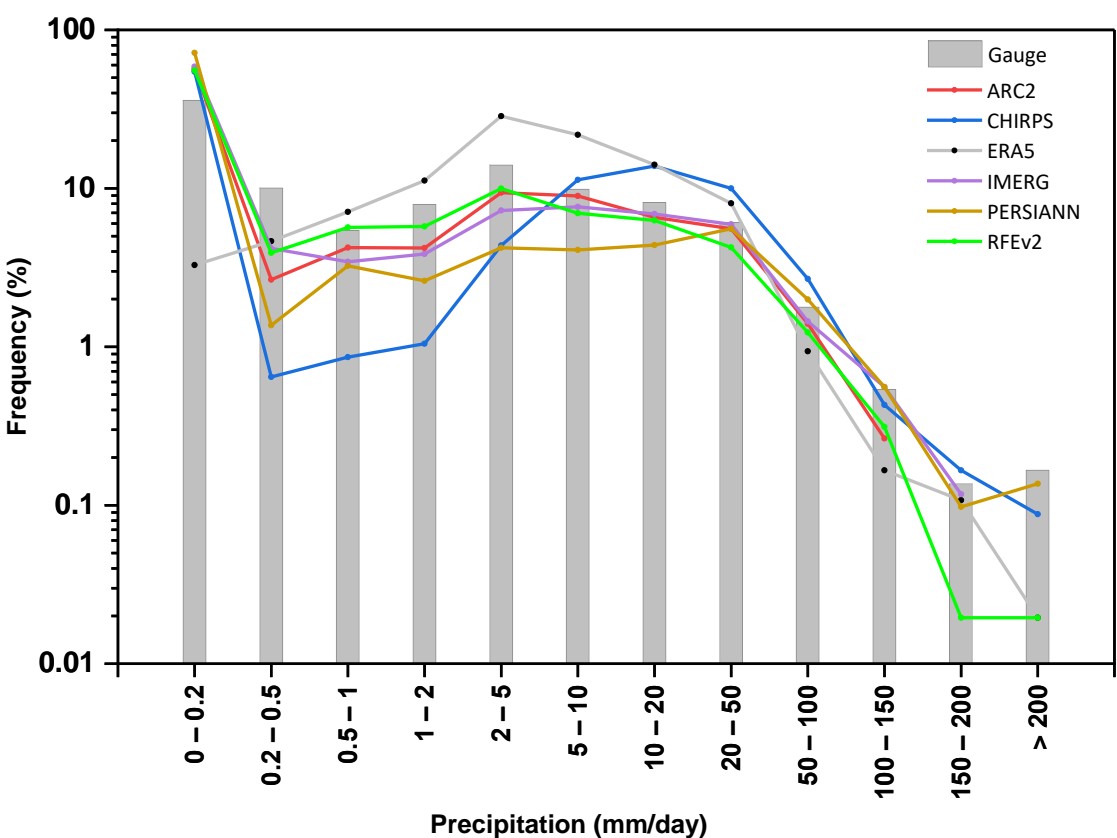

**Figure 5.** Probability distribution function (PDF) of daily rainfall intensities based on GPP and rain gauge data from September 2018 to August 2020. Note logarithmic (Log10) scale for Y axis.

### 3.4. Precipitation Detection Ability

Figure 6 depicts the rainfall detection ability (POD, FAR, CSI) of the various GPPs. The results show that the POD values of ERA5 are highest among all products for the precipitation classes between 0 and 5 mm/day but are among the lowest for rainfall classes > 20 mm. As a matter of fact, the POD of ERA5 decreases steadily with increasing daily rainfall amount. The POD of ARC2, IMERG, RFEv2, and CHIRPS are similar (approximately 0.5 to 0.6) for the precipitation classes < 1 mm/day. However, the POD of IMERG remains rather constant in the range of 0.2–100 mm/day, whereas the PODs of ARC2 and RFEv2 decrease steadily. The POD of CHIRPS first increases slightly up to 5 mm/day and then decreases thereafter. PERSIANN performs worst of all GPPs for the lowest precipitation classes, but the POD tends to increase with increasing rainfall. For precipitation classes > 100 mm/day, only the PERSIANN product has a high POD value, while the PODs for all other GPPs tend towards 0.

The FARs of all GPPs increase steadily between 0.2 and 20 mm/day. For rainfall classes ≤ 100 mm/day, IMERG shows similar or better performance than all other products. For daily rainfall > 150 mm, only PERSIANN has a low FAR.

Based on the CSI value, the ERA5 product performs best in detecting precipitation in the 0.2–2 mm/day precipitation range. IMERG performs best in detecting rainfall in the range of 2–100 mm/day. Only PERSIANN performs well for rainfall > 150 mm/day. Both IMERG and PERSIANN show rather constant performance in terms of CSI in the range of 0.2–100 mm, but IMERG outperforms PERSIANN in this range.

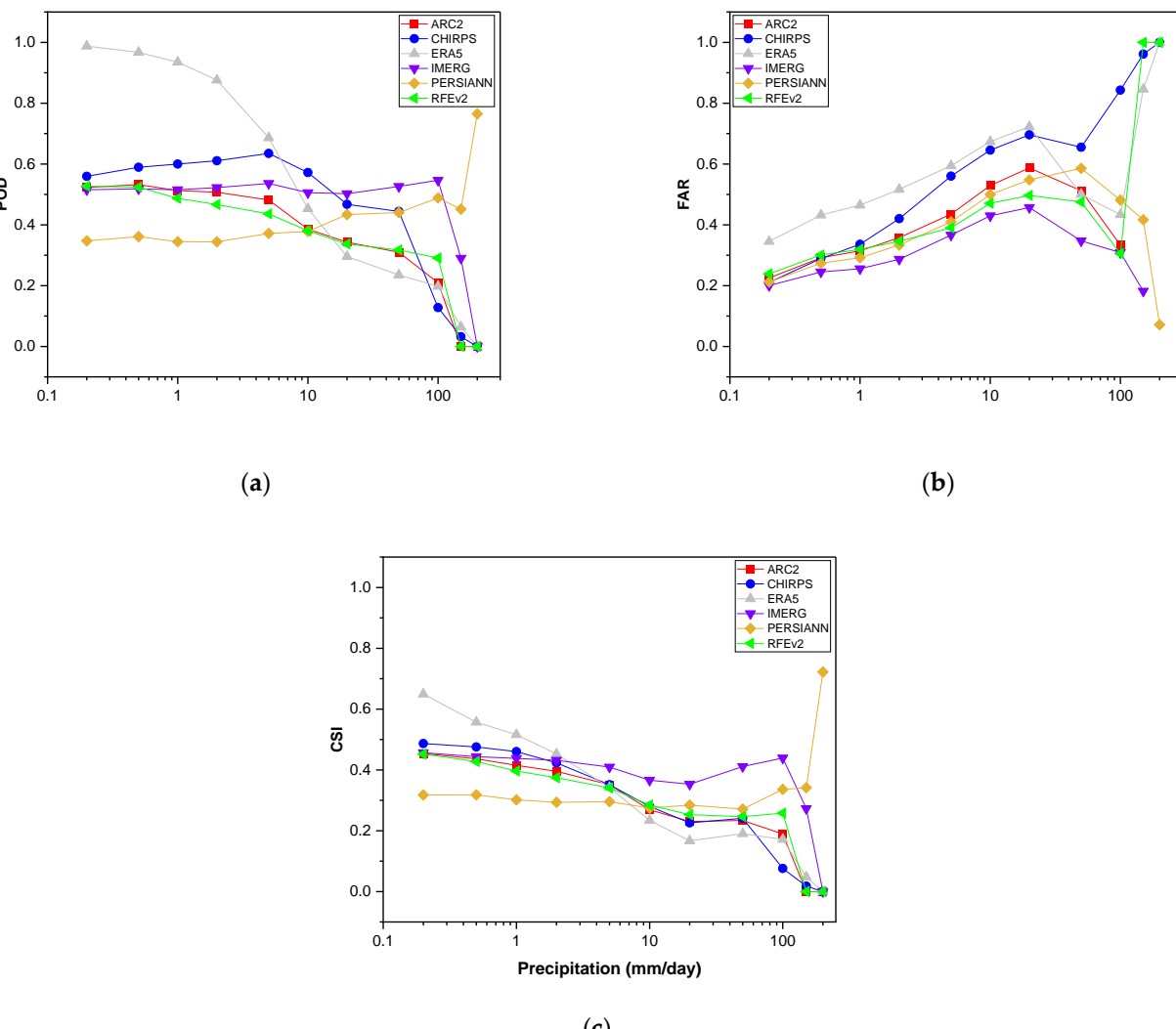

**Figure 6.** Precipitation detection ability of the six GPPs based on daily rainfall: (**a**) Probability of detection (POD), (**b**) False alarm ratio (FAR), (**c**) Critical success index (CSI). Ground-based rain gauge data from September 2018 to August 2020 are used as reference. Note logarithmic (Log10) scale for X axis.

*3.5. Different Time Scales Assessment (Hourly to Yearly)*

For the different time scales, only the IMERG product was evaluated, using the point-gridded approach, given that this product appeared to perform best at the daily time scale (§ 3.2). Figure 7 depicts the evaluation of IMERG at different time scales (hourly to yearly) against the gauge data over the Ankavia catchment. The coefficient of determination increases with increasing aggregation time scales from hourly to yearly. Specifically, IMERG exhibits good correlation at the yearly time scale ($R^2$ = 0.97; Figure 7a) and at monthly time scale ($R^2$ = 0.87; Figure 7b), and reasonable correlation at the daily assessment ($R^2$ = 0.65; Figure 7c). The correlation is poor at the hourly time scale ($R^2$ = 0.20; Figure 7d). Especially at the daily time scale, there is a tendency to underestimate the events > 150 mm (Figure 7c). IMERG also tends to underestimate yearly and, to a lesser extent, monthly rainfall. Additionally, it is apparent that the variability in yearly (Figure 7a) and monthly (Figure 7e) rainfall across the watershed is greater than the variability in IMERG rainfall data, especially for the high-rainfall months.

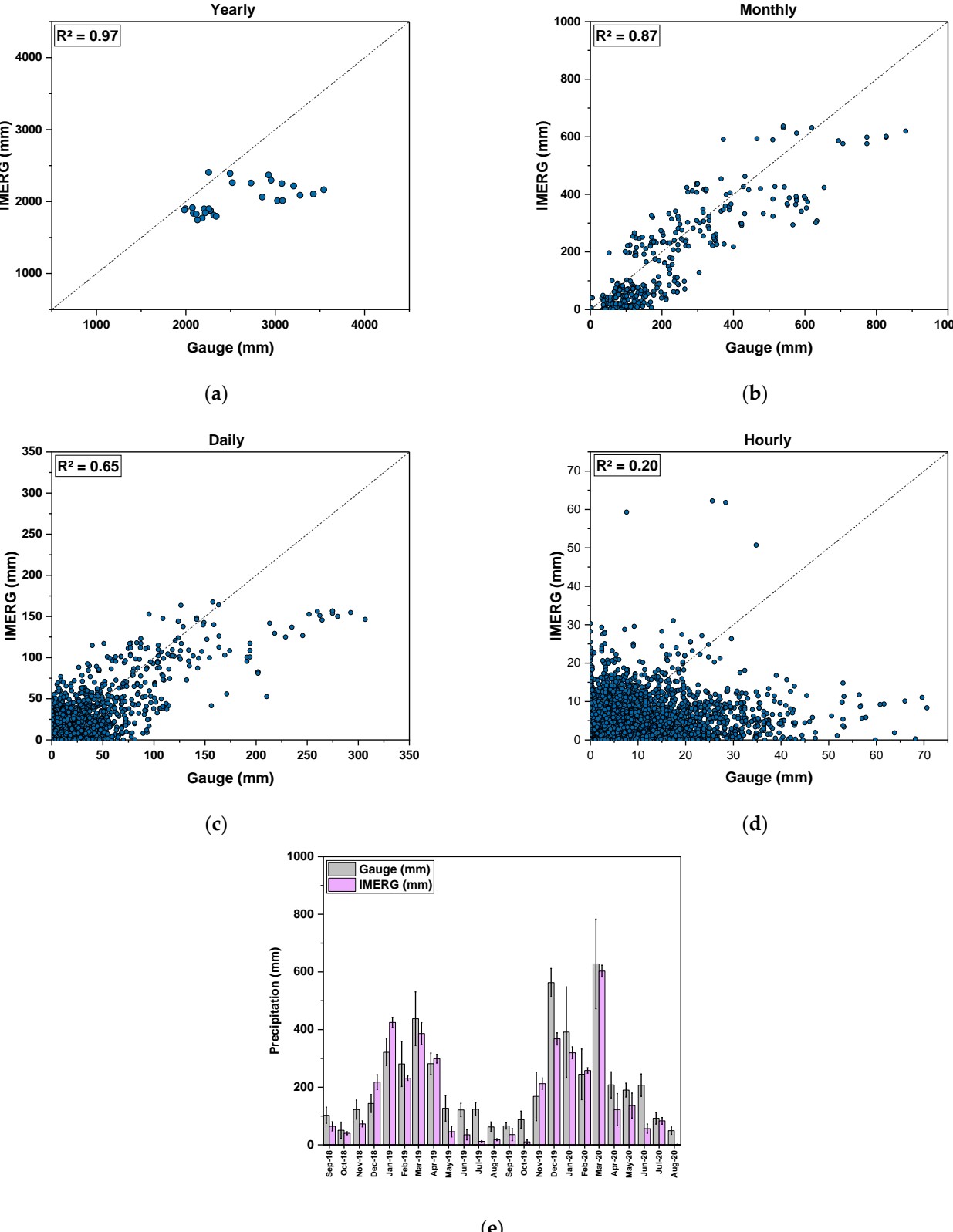

**Figure 7.** Comparison of ground-based precipitation data with IMERG at different time scales: (**a**) yearly, (**b**) monthly, (**c**) daily, (**d**) hourly, (**e**) barplot of monthly rainfall with standard deviation. $R^2$ value is determined for regression passing through (0,0).

### 3.6. Event Scale Assessment

Figure 8 shows the PDF plots for the rainfall event depths, durations, and intensities over the 2-year period. Overall, the IMERG precipitation product presents a good agreement with rain gauge data in terms of duration (Figure 8a) and depth (Figure 8b). However, the rainfall intensities between 0–5 mm/h are underestimated, while the 5–10 mm/h rainfall intensity class is largely overestimated (Figure 8c).

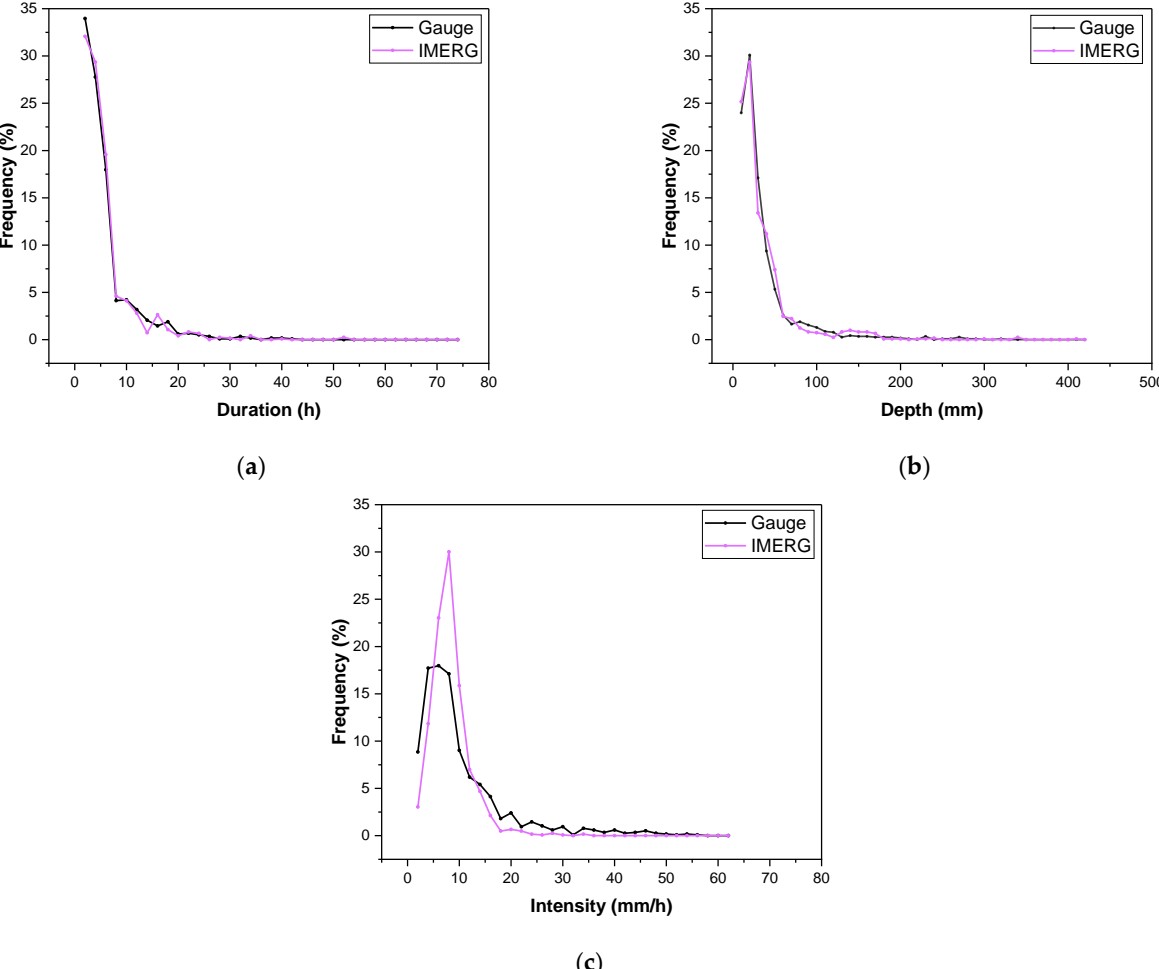

(**a**)                                                                                      (**b**)

(**c**)

**Figure 8.** Probability distribution function of (**a**) rainfall duration, (**b**) rainfall depth, and (**c**) rainfall intensity at event time scale for ground-based rain gauges and IMERG data from September 2018 to August 2020.

## 4. Discussion

For the validation of the GPP data using ground-based rain gauges, the point-gridded approach performs similarly or better than the point-to-grid approach (Figure 3). Since the grid cell centroids rarely coincide with the rain gauge position, the point-to-grid method is subject to greater inaccuracy [60,61]. This inaccuracy grows in proportion to the spatial resolution of the GPP [62]. In contrast, the point-gridded approach reduces this mismatch problem between gauge data and GPP data while preserving the resolution of the gridded data. Nevertheless, the point-gridded technique is rarely studied [62].

At the daily time scale, among the six GPPs, IMERG performed best at our study location (Figure 3). Its overall performance is reasonably good despite the small number of available rain gauges used for the Global Precipitation Climatology Centre (GPCC) in Africa, especially in Madagascar [53], resulting in biased precipitation data at rain gauge and catchment scale. Several previous studies have also underlined that IMERG outperforms other GPPs in tropical areas [20,63,64]. This has been attributed to the GPM

Microwave Imager (GMI) and the ability of the Ku/Ka-band Dual-frequency Precipitation Radar (DPR) to capture precipitation more effectively than the IR sensors and/or direct PM technologies used by other GPPs [53]. Specifically, the GMI instrument is a conically scanning, multi-channel microwave radiometer with 13 channels ranging in frequency from 10 GHz to 183 GHz [53]. The GMI employs a set of frequencies that have been refined over the last two decades to recover most ranges of precipitation, with the polarization difference of each channel serving as an indicator of optical thickness, water content, and precipitation systems [53]. Additionally, RFEv2, ARC2, and PERSIANN all show satisfactory performance (Figure 3). In contrast, CHIRPS and ERA5 perform poorly in the daily assessments. Some factors, such as gauge calibration and the sources of products play significant roles in the performance of GPPs [62]. In general, satellite-based precipitation products are currently more appropriate for meteorological applications than model-based products (e.g., ERA5) in tropical areas, especially over regions with extreme precipitation events. Nevertheless, reanalysis products such as ERA5 could have significant advantages in temperate regions or for research focusing on occurrence detectability at finer resolutions over high-latitude areas [49]. Therefore, the fact that some products (ARC2, IMERG, PERSIANN, RFEv2) are gauge calibrated and satellite-based likely explains their better performance compared to CHIRPS and ERA5.

Table 3 summarizes the results from various studies that evaluated the performance of IMERG at a daily time scale in broadly similar climatic environments to the present study (humid tropical environment). Note that almost all of those studies used a point-to-grid or grid-to-grid approach; hence, the performance assessments are largely influenced by the density of the gauge network taken as a reference [63,65,66]. In terms of correlation (CC), the results from our study are among the highest reported so far, similar to [63]. RMSE and MAE values in the present study are within the range of those reported previously. FAR values are also within the range of previously reported values, whereas POD and CSI are somewhat lower in the present study compared to previous studies. Nevertheless, IMERG's fairly high POD, CSI, and low FAR suggest a good detection capability for daily rainfall, particularly in the range of 0–100 mm/day (Figure 6), even though some rainy days are still being missed. Precipitation events estimated by the GPP may not be detected by the gauges as it might rain at other locations within the grid-cell area. Furthermore, given their spatial resolution, GPPs will be less sensitive to short-range variations in rainfall, which may explain the somewhat poorer performance of IMERG in the present study, given that many gauges are separated from each other by less than 10 km (i.e., roughly 0.1°, the spatial resolution of an IMERG grid box). Additionally, many factors could influence this variation of performance across studies, including the density of the rain gauge network and especially the validation technique [25,63,67].

**Table 3.** Summary of IMERG assessment studies in tropical environments at a daily time scale.

| References/ Study Area | Study Period/ Number of Rain Gauges for Validation | Validation Approach | CC or $\sqrt{R^2}$ | RMSE mm/Day | MAE mm/Day | POD | FAR | CSI |
|---|---|---|---|---|---|---|---|---|
| [25]/East Africa | 2000–2018/36 | grid-to-grid | 0.41 | 12.4 | 7.6 | 0.88 | | |
| [66]/East Africa | 2014/37 | grid-to-grid | 0.53 | | | 0.87 | 0.04 | |
| [68]/Singapore | 2014–2016/48 | grid-to-grid | 0.53 | 11.83 | | 0.78 | 0.28 | 0.60 |
| [63]/Philippines | 2014–2017/55 | grid-to-grid | 0.81 | 5.66 | 3.74 | | | |
| [64]/Bali | 2015–2017/27 | point-to-grid | 0.32 | 17.19 | | 0.84 | 0.54 | 0.44 |
| [69]/Vietnam | 2014–2016 53 | grid-to-grid | 0.58 | | | 0.73 | 0.22 | 0.61 |
| [70]/Malaysia | 2014–2016/31 | point-to-grid | 0.54 | 14.93 | | 0.89 | 0.20 | 0.73 |
| [62]/Mexico | 2014–2015/99 | point-gridded | 0.54 | 7.93 | | 0.2–0.6 | 0.2–0.6 | 0.2–0.8 |
| Ankavia | 2018–2020/14 | point-gridded | 0.80 | 12 | 5.5 | 0.5–0.6 | 0.2–0.4 | 0.4–0.5 |

According to our results, the performance of IMERG is rather uniform across the watershed, i.e., there is no evidence of a spatial trend of the statistical metrics (Figure 4). Therefore, the negative effect of the topography, which often alters the performance of the precipitation satellites, is not apparent in our study area. Nevertheless, our results are consistent with previous findings for stations located in mid- and low-altitude, with relatively mild and wet climates [67,71,72]. Indeed, 13 of the 14 rain gauge stations in the Ankavia catchment are located between 14 and 300 m a.m.s.l.

The PDF analyses underline that most of the GPPs show the same distribution as the gauges except for the ERA5 data and, to a lesser extent, the CHIRPS data. Overall, the results reveal an overestimation for the 0–0.2 mm/day precipitation class and an underestimation of the >0.2 mm/day classes (Figure 5). Other studies have, however, reported that IMERG slightly overestimated the frequency of rainfall events between 1 and 50 mm/day [64,68,70]. In addition, the findings also indicate that some GPPs underestimate the precipitation classes >150 mm/day. The poor performance of GPPs at detecting extreme events was also reported in other assessment studies in tropical river basins [23,68]. Specifically, since gridded products contain spatially-averaged rainfall values, larger grid size ($0.1 \times 0.1°$) products are more likely to smooth out the extreme rainfall values (>150 mm/day), which are especially associated with short-duration events with limited spatial extent [69,73]. In contrast, CHIRPS and PERSIANN outperform all other GPPs in this range (>150 mm/day). Their capacities to better represent very high-intensity rainfall could be due not only to their smaller grid size ($0.05°$ and $0.04°$, respectively) but also to their ability to categorize cloud-patch features based on height, areal extent, and variability of texture estimated from satellite imagery [46,55]. These classifications aid in the assignment of rainfall values to pixels within each cloud based on a predefined curve that describes the link between rain rate and brightness temperature [55].

With respect to the time scale of integration, the correlation between ground-based data and IMERG data improves with increasing summing time scales from hourly to yearly, which is in line with others' findings [25,64,68,70]. However, the strength of the correlation at a yearly timescale is constrained by the short duration of the study period (2 years). In addition, IMERG data have difficulty reproducing the spatial variability of rainfall within the catchment (Figure 7a,b,e). This may at least partly result from the spatial smoothing inherent to GPPs, daily extreme events not identified by the satellite (Figure 5), as well as the small number of rain gauges used as the bias correction for IMERG [53]. At the event time scale, there is good agreement in terms of PDF of duration and depth (Figure 8), which is consistent with the results of [59] in Brazil. Finally, the poor performance of IMERG on an hourly scale has also been reported by other studies [74,75]. This is particularly due to the inferior performance at locations where the estimate is derived by morphing, which occurs when no overpasses by any of the passive microwave instruments in the GPM constellation are available at a specific half-hour [53]. This is also due to the temporal resolution of inputs (>1 h), cited above (Section 2.3.4), used to calculate the IMERG product [53,74].

In this paper, we only focused on IMERG for the assessment at different time scales due to its better performance at a daily scale. However, it is important to note that the poor performance of some products at a finer scale does not necessarily suggest that they will be bad at longer time scales. As a matter of fact, the accuracy of gridded precipitation products increases as the time of accumulation increases [25,64,76].

## 5. Conclusions

In this study, we performed the first assessment of six gridded precipitation products (ARC2, CHIRPS, ERA5, IMERG, PERSIANN, RFEv2) over the Ankavia watershed in Madagascar, for a common period from September 2018 to August 2020, with 14 rainfall gauges taken as reference. The main findings of the study can be summarized as follows:

- The point gridded approach is better suited than the point-to-grid approach in terms of continuous statistical metrics to evaluate gridded precipitation products against rain gauge data;

- At a daily scale, IMERG outperforms all other tested gridded precipitation products, followed by RFEv2 and ARC2;
- GPPs tend to overestimate the 0–0.2 mm/day rainfall class but underestimate the >0.2 mm/day ranges. Only GPPs with smaller grid sizes (0.04°, 0.05°) accurately estimate the >150 mm/day precipitation class;
- IMERG is shown to perform well in detecting rain events up to 100 mm/day but is surpassed by PERSIANN in detecting rain events larger than 150 mm/day. Nevertheless, a substantial proportion of rainy days are not correctly estimated by IMERG;
- IMERG shows good performance at monthly, daily, and event time scales in our case study; nevertheless, its capacity to reproduce spatial variability of rainfall is very subpar at the catchment scale.

Overall, this GPP assessment study in northeastern Madagascar provides evidence that the IMERG v06 final precipitation datasets perform satisfactorily when compared to rain gauge time series using the point-gridded technique at a daily time scale. In addition, the level of performance is fairly constant across a broad range of daily rainfall values, except for extreme events. Therefore, IMERG is the most reliable for estimating rainfall characteristics in this region. However, the product should be used with caution for hazard and flood assessment, given its limitations for extreme rainfall events.

**Author Contributions:** Conceptualization, Z.R. and C.L.B.; Formal analysis, Z.R. and C.L.B.; Funding acquisition, M.V.; Investigation, Z.R.; Methodology, Z.R. and C.L.B.; Project administration, M.V.; Software, Z.R.; Supervision, A.R. and C.L.B.; Validation, Z.R., A.R. and C.L.B.; Visualization, Z.R., A.R., F.J., M.V. and C.L.B.; Writing—original draft, Z.R.; Writing—review and editing, Z.R., F.J., M.V. and C.L.B. All authors have read and agreed to the published version of the manuscript.

**Funding:** This research and APC fees are funded by the Belgian Académie de Recherche d'Enseignement Supérieur (ARES-CCD: www.ares-ac.be) (accessed on 26 February 2022), through the 2017 Research Project for Development (PRD) in Madagascar, named: Renforcement des Capacités en Gestion Integrée des Ressources en Eau (GIRE-SAVA). More information on: https://www.ares-ac.be/fr/cooperation-au-developpement/pays-projets/projets-dans-le-monde/item/162-prd-renforcement-des-capacites-en-gestion-integree-des-ressources-en-eau-de-la-region-sava-gire-sava (accessed on 14 April 2022).

**Data Availability Statement:** The observed rain-gauge precipitation data used in this study are from the GIRE SAVA project (available on request).

**Acknowledgments:** The authors acknowledge all those who gave their support for the accomplishment of this study. Our special thanks go to the south coordinator Joseph Benitsiafantoka (2017–2021), Christophe Manjaribe (since 2021, and their respective staff in the Centre Universitaire Régional de la SAVA (CURSA). We also acknowledge the contribution of the members of the Sambava regional direction of water, especially Tsirinasy (2017–2021), Carlos Totomalaza (since 2021), the heads of the surveyed communities, and the survey participants (technicians and research assistants).

**Conflicts of Interest:** The authors declare no conflict of interest. The funders had no role in the design of the study; in the collection, analyses, or interpretation of data; in the writing of the manuscript, or in the decision to publish the results.

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
