# Peer review of "Reliability of Gridded Precipitation Products for Water Management Studies: The Case of the Ankavia River Basin in Madagascar"

_remotesensing, doi:10.3390/rs14163940_

Round 1

Reviewer 1 Report

See attached file.

Author Response

We would like to thank you for these relevant comments.

Please see the attachment for the response.

Best regards.

Reviewer 2 Report

  See attached file.

Author Response

(The authors gave the same response as above.)

Reviewer 3 Report

Review of “Reliability of satellite-based rainfall products for water management studies: the case of the Ankavia river basin in Madagascar” by Z. Ramahaimandimby, et al., submitted to Remote Sensing

July 21, 2022

Recommendation: Accept after minor revision

General Comments

This paper evaluates six satellite-based rainfall products over the Ankavia catchment in Madagascar over a two-year period.  In doing so, the authors recognize the danger in comparing point measurements from gauges with gridded precipitation estimates.  They address this via a “point-gridded” approach that should generally yield a more robust validation.  IMERG was found to perform the best and was examined more closely at various time scales.

Overall, the paper is very well-written and easy to follow.  My main concern is whether a sample size of just fourteen gauges over two years is enough to adequately represent the performance of the various satellite-based products.

Major Comments

1. It is important to specify which version of a satellite algorithm you are examining.  PERSIANN (abstract/line 22 and line 61) should be PERSIANN-CCS, but this is not made clear until line 245.  IMERG should be specified as IMERG V06 Final, but this is not made clear until line 242.  Further the mentions of the phrase “Day 1” in regard to IMERG (lines 239-241) should be deleted, as this term was used only in connection with the at-GPM-launch version of IMERG back in 2014.

2. The difficulty IMERG has in retrieving extreme rainfall rates is a known issue that is primarily due to limits imposed in V06 of 120 mm/hr for microwave retrievals and 50 mm/hr for IR retrievals.  These upper limits are being increased to 200 mm/hr in the soon-to-be-released IMERG V07.  Meanwhile, the manuscript is not consistent in its mention of the upper limit of usability in IMERG V06: lines 31 and 528 say 150 mm/day (although Fig. 6c perhaps indicates closer to 100, as mentioned in line 468), and lines 40 and 373 say 200.

3. As mentioned above, two years is a short record.  Was only this period available in the gauge record?  After all, line 178 says in reference to the satellite products, “these products were chosen based on the availability of long time series”.  This issue is exacerbated in Fig. 7a, where the scatterplot for IMERG yearly apparently contains only 28 data points (14 gauge stations * 2 years).

4. The explanation for the relatively poor performance of IMERG at hourly scale (lines 513-514) can better be described as due to the inferior performance at locations where the estimate is derived by morphing, which occurs when no overpasses by any of the passive microwave instruments in the GPM constellation were available at a particular half-hour.

Minor Comments

1. Line 44: change “is” to “are”.

2. Line 62: add “, among others.” at the end of the sentence to indicate that there are other products (e.g., CMORPH, GSMaP) not examined here.

3. Line 63: “<” should be less than or equal to 0.1.

4. Line 86: “IMERG” should be “GPM”, but actually, there is no need to spell out the full acronym here since you have already done so at line 60.  In fact, all six products are fully spelled out in the paragraph beginning on line 56, so there is no need to use their full names again in Section 2.3.

5. Line 89: change “predict” to “retrieve” or “agree with”.

6. Line 101: how is “explored” different from “quantitatively analyzed”?

7. Line 112: “PERSSIAN” is a misspelling.  Further, it should say “PERSIANN-CCS”, perhaps adding “hereafter PERSIANN” so that you do not have to add “-CCS” every time.

8. Line 138: “has the highest rainfall”.  Do you mean highest in the country?

9. Lines 140-141: spell out “approximately”.

10. Line 173: change “2-years” to “two-year”.

11. Lines 239-240: the latencies for Early and Late are 4 and 14 hours, respectively.

12. Line 264: this should be “Special Sensor Microwave Imager/Sounder (SSMIS)”.

13. Line 268: I double-checked the website address, and it is “.RFEv2” not “.REFv2”.  I was going to point out on line 270 that “RFE” should be “REF2”, but now I am confused.  The cited website calls the product “RFEv2”, yet this manuscript calls it “REF2”.

14. Table 1: for IMERG, capitalize “Precipitation Measurement”.

15. Line 297: “differents” should be “different”.

16. Line 360: I don’t understand what you are trying to say.  Perhaps you mean “pattern” and not “trend”?  However, you have somewhat already said this two sentences before.

17. Line 369: It appears that CHIRPS also overestimates between 0-0.2 mm/day.

18. Line 378: “rainfal” should be “rainfall”.

19. Line 405: “differents” should be “different”.

20. Line 442: spell out the GPCC acronym.

21. Line 457: change “may explain” to “likely largely explains”.

22. Line 473: after “less than 10 km”, add “(i.e., roughly 0.1°, the spatial resolution of an IMERG grid box)”.

23. Line 504: add an apostrophe after “others”.

24. Line 521: “precipitations” should be “precipitation”.

25. Line 523: change “the IMERG” to “IMERG” and “the REF2” to “REF2”.

26. Line 530: change “predicted” to “retrieved”.

27. Line 532: change “show” to “shows”.

28. Line 675: in reference 52, change “2015, 612” to “612, 2015”.  But actually, a much more up-to-date reference for IMERG V06 is:  Huffman, G.J., Bolvin, D.T., Nelkin, E.J.,, Tan, J., 2020: Integrated Multi-satellitE Retrievals for GPM (IMERG) Technical Documentation. NASA/GSFC Code 612, 2020, 83 pp.

Author Response

(The authors gave the same response as above.)

Round 2

Reviewer 1 Report

The authors have addressed my comments. I recommend acceptance of the manuscript.

Author Response

We would like to thank the reviewer for your careful reading of the revised manuscript and also for your appreciation of the quality of the work. 

We are sincerely grateful for your approval.

Reviewer 2 Report

Authors aggressed all my previous comments and they made several additional suggested changes. The manuscript was sufficiently improved. Following suggestions may be considered before the manuscript will be published.

Line 91: Would be essential to explain meaning of GPM = Global Precipitation Measurement

Line 94: It looks that “retrieve” is not proper term in this sentence. Better word would be to use “estimate”.

Line 570: Similarly, I suggest replacing “retrieved” with “estimated”

Author Response

Please see the responses to the comments in the attachment.

Reviewer 3 Report

Recommendation: Accept after minor revision

General Comments

The authors have addressed the points I raised in my initial review.  I have only a few minor comments on the revised manuscript.

Minor Comments

1. Line 265: change “in following several steps, which consist to” to “in several steps, which consist of”.

2. Line 349: change “observed precipitations” to “observed precipitation occurrences”.

3. Line 394: change “precipitations” to “precipitation”.

4. Line 395: I don’t think “evaluate” is quite the right word.  I think “retrieve” would be better, both here and in place of “estimate” with regard to ARC2 on line 394 and in place of “estimate” with regard to RFE on line 396.

5. Reference 52 does not appear to have been updated.  Again, the appropriate citation is:  Huffman, G.J., Bolvin, D.T., Nelkin, E.J., Tan, J., 2020: Integrated Multi-satellite Retrievals for GPM (IMERG) Technical Documentation. NASA/GSFC Code 612, 2020, 83 pp.

Author Response

(The authors gave the same response as above.)
